# Imaging radiation dose in breast radiotherapy by X-ray CT calibration of Cherenkov light

R. L. Hachadorian [1], P. Bruza [1], M. Jermyn[1,2], D. J. Gladstone [1,3,4], B. W. Pogue [1,2,3,4] & L. A. Jarvis [3,4]✉

Imaging Cherenkov emission during radiation therapy cancer treatments can provide a real-time, non-contact sampling of the entire dose field. The emitted Cherenkov signal generated is proportional to deposited dose, however, it is affected by attenuation from the intrinsic tissue optical properties of the patient, which in breast, ranges from primarily adipose to fibroglandular tissue. Patients being treated with whole-breast X-ray radiotherapy ($n = 13$) were imaged for 108 total fractions, to establish correction factors from the linear relationships between Cherenkov light and CT number (HU). This study elucidates this relationship in vivo, and a correction factor approach is used to scale each image to improve the linear correlation between Cherenkov emission intensity and dose ($R^2_{6X} = 0.85$, $R^2_{10X} = 0.95$). This study provides a major step towards direct quantitative radiation dose imaging in humans by utilizing non-contact camera sensing of Cherenkov emission during the radiation therapy treatment.

[1] Thayer School of Engineering, Dartmouth College, 14 Engineering Dr., Hanover, NH 03755, USA. [2] DoseOptics LLC, 16 Cavendish Ct., Lebanon, NH 03766, USA. [3] Geisel School of Medicine, Dartmouth College, 1 Rope Ferry Road, Hanover, NH 03755, USA. [4] Norris Cotton Cancer Center at Dartmouth Hitchcock Medical Center, 1 Medical Center Dr., Lebanon, NH 03756, USA. ✉email: Lesley.A.Jarvis@Hitchcock.org

Since the discovery of X-rays, they have been utilized ubiquitously for medical diagnostic imaging and therapy. Yet despite nearly unlimited ability to measure the transmission of the beam through tissue, there is not currently a way quantify the radiation dose deposited directly in tissue. Techniques employed to measure radiation deposition, usually based upon gas ionization or charge separation in devices[1], have always been external to the body, and noninvasive or remote ways to measure the absorbed dose in tissue would add a critically important layer to patient safety and management. Recently, the concept of estimating subsurface dose was established by imaging the distribution and intensity of Cherenkov light emission during radiation therapy[2]. This light is a low-intensity, broad-spectrum signal that is emitted along the path of high-energy, dose-depositing electrons. Cherenkov imaging (typical setup shown in Fig. 1) has enabled both the relative mapping of the radiotherapy beam shape as it is delivered in real time, and has illustrated that the time-integrated Cherenkov field outline matches the planned dose delivery[3] (for details on the mechanisms of Cherenkov imaging and iCMOS camera function, see Methods). In this study, a methodology to quantify radiation dose with Cherenkov light has been examined and demonstrated using breast irradiation imaging and diagnostic imaging data from X-ray CT (computed tomography) scans, available from these patients.

Until Cherenkov imaging systems became available for research use in 2014, it was not possible to see the beam treating the patient, nor if the deposited dose was consistent each treatment fraction. Examples of the therapist's ability to visualize the treatment in real time can reveal if the patient's arm, chin, or contralateral breast were problematically not clear of the treating beam. One motivation for the proposed work is to provide a simple tool that allows the radiotherapy team to see the radiation dose deposition as it happens in the field of the beam, directly on the patient. Additionally electronic image capture allows for recording each imaging session, such that if small errors occur, then measures can be taken to fix a systematic issue as opposed to never knowing they existed otherwise. Cherenkov imaging of radiotherapy dose is the first imaging modality that could provide whole-field, real-time, dose imaging in radiation therapy.

This study presents a major step in a multi-part solution to the most limiting aspect of in vivo dosimetry. This limitation surrounds the attenuation that diffuse Cherenkov light undergoes as it propagates in tissue. In an unattenuating medium (a medium lacking optical absorbers, such as water), Cherenkov light will serve as a direct surrogate for absolute radiation dose[4], but this attenuation in tissue alters the linearity between the radiation dose and the observed Cherenkov signal exiting the skin surface. This attenuation is also nonlinear due to patient-specific differences in tissue, such as adipose versus fibroglandular content, blood content, radiation burn (erythema)[5], musculature, or skin color; all of which ultimately yield substantial variation in optical absorption and scattering[6,7]. It is well known that various tissue types have specific blood volumes and optical scattering[8–10], thus it seems plausible that internal tissue characterization using X-ray CT attenuation values (HU, Hounsfield units) might be used as a way to calibrate for the heterogeneous optical transport of subsurface Cherenkov photons. This particularly relevant correction takes advantage of the ability to measure breast tissue densities across a spectrum of compositions from adipose to fibroglandular, readily separated by CT number[11–13], and that these two tissue types are known to have very different optical properties[10,14,15].

Another limitation in Cherenkov imaging is that only near-surface dosimetry information is yielded, whereas the volumetric region of interest is usually throughout the breast volume. Despite dose information being only available at the surface, Cherenkov imaging is still a unique modality with the capacity to provide real-time tracking of large areas of tissue undergoing treatment; considerably broader than traditional surface dosimetry methods,

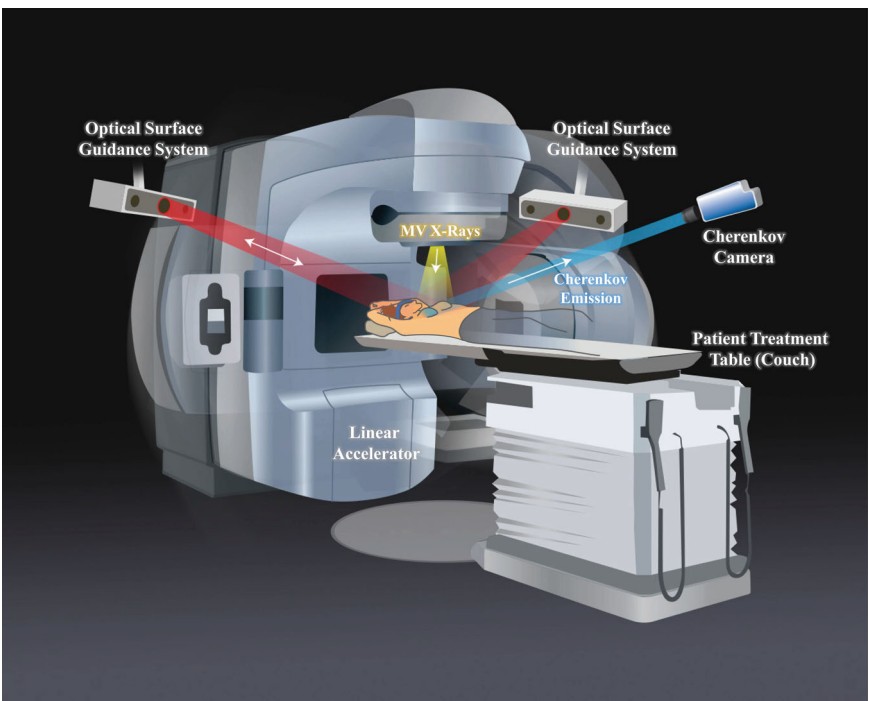

**Fig. 1 Study setup and patient positioning.** The Cherenkov camera and optical surface guidance projectors and cameras were fixed to the ceiling. The linear accelerator (linac) gantry rotates to each beam position and remains stationary for delivery of each field. When the linac beam of X-rays (yellow) is incident upon the tissue, Cherenkov light is emitted isotropically from within. Some of this light is detected by the Cherenkov camera (blue). The camera intensifier is triggered on during only the linac pulses, thereby suppressing ambient light interference. The optical surface guidance system cameras are used to set up the patient and ensure correct alignment by casting a red, diffuse light pattern onto the patient, and tracking respective movement.

which are limited to individual locations on the skin surfaces[16]. Point measurements have traditionally been used to assess the accuracy of execution of the entire treatment field, though in contrast, the Cherenkov light signal is emitted everywhere dose is being actively deposited, which allows the therapy team to verify dose anywhere within the field. Additionally, changes in dose at the surface can potentially be used to infer changes in dose deeper in the tissue[17]. However, the key factor in this work remains that the calibration of Cherenkov light intensity to absolute dose would be the first step in a potential paradigm shift in how radiotherapy delivery is visualized and verified. Further, therapeutic X-rays are the most common treatment modality for cancer, with over 50% of all patients receiving it, it is estimated that approximately 11 million women being treated for invasive breast cancer in the United States alone could benefit from to continued exploration of imaging in vivo dosimetry.

In this study, we present a method of calibrating the Cherenkov emission to absolute dose using X-ray radiodensity for macroscopic tissue types following the workflow in Fig. 2. This schematic shows the methodology proposed for correcting cumulative Cherenkov light fields (Fig. 2a) to quantitative absolute dose maps (Fig. 2b). This is carried out by generating corrections for absorbed light using a linear correlation between these dose-normalized Cherenkov intensities (dividing Fig. 2a by co-registered Fig. 2b) and the average HU per patient (Fig. 2c). CT number (HU) can be readily obtained from standard clinical equipment, namely treatment planning CT scans[18–20], which is an important consideration for time and workflow efficiency. Correcting the Cherenkov intensities using this model is shown to notably reduce the macroscopic variations between observed Cherenkov emission and radiation dose. In the current cohort of 108 imaging sessions of fractionated breast cancer radiotherapy, we present a comprehensive case study of quantitative remote optical radiation dosimetry in human tissues.

## Results

**Patient imaging.** Informed consent was obtained from each patient enrolled in this IRB-approved study. Representative Cherenkov images from each of the 13 patients are shown in Fig. 3. Here, the Cherenkov signal is shown for one cumulative fraction in false color, overlaid on the co-acquired, background, room light image. Patients were irradiated with the standard of care for whole-breast irradiation, using two opposing beams from the right posterior oblique (RPO) side and the left anterior side (LAO). Patients 3 (Fig. 3c), 7 (Fig. 3g), 8 (Fig. 3h), 10 (Fig. 3j), and 13 (Fig. 3m) were all treated using two sets of opposing

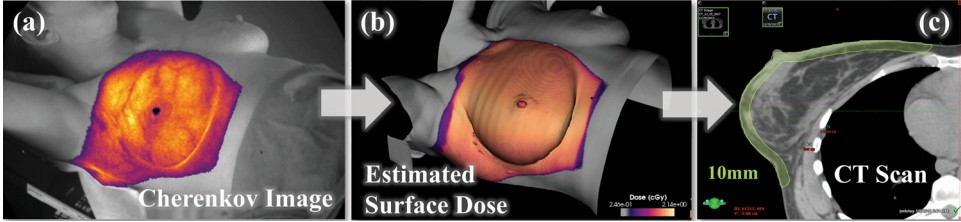

**Fig. 2 Model to build correction factors.** Cherenkov frames are recorded throughout a fraction of treatment and summed into a cumulative image (**a**). Each beam was then separated out by beam energy and gantry angle. Each Cherenkov image is divided by the respective co-registered surface dose image (**b**) rendered in C-Dose software from the treatment plan (Supplementary Fig. 1). The patient planning CT scan is then sampled down up to 10 mm for an average CT attenuation number (HU) (**c**), done in the treatment planning software by creating a structure (contour pictured). A correction factor is calculated from both average CT attenuation and Cherenkov normalized by dose. In an ideal scenario, the corrected Cherenkov image corrects for tissue optical properties and shares a higher metric of uniformity with the surface dose image (**b**).

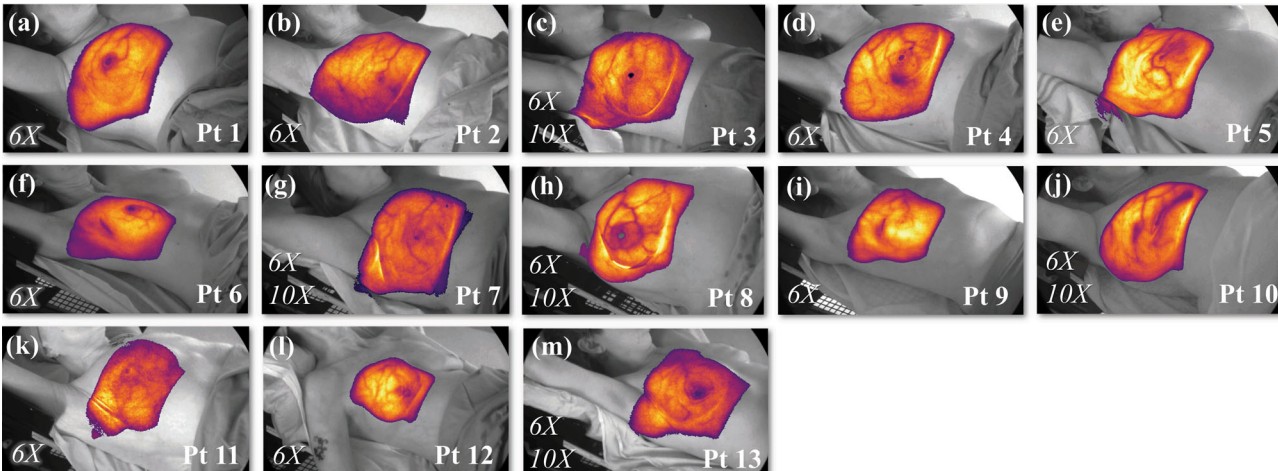

**Fig. 3 Cherenkov image data from 13 whole-breast radiotherapy patients.** Patients 1 (**a**), 2 (**b**), 4 (**d**), 5 (**e**), 6 (**f**), 9 (**i**), 11 (**k**), and 12 (**l**) had prescribed radiation treatments using only 6 MV energy beams (6X), and treatments for Patients 3 (**c**), 7 (**g**), 8 (**h**), 10 (**j**), and 13 (**m**) included both 6 and 10 MV beams (6X/10X), indicated in the bottom left corner for each patient thumbnail. The recorded Cherenkov emission is overlain in color on the recorded background image (grayscale), which is captured in real time. In (**h**), a small patch bolus was used as buildup material over the surgical scar (right lateral, region of higher intensity in the lateral mammary fold)[29]. In each thumbnail, Cherenkov fields have undergone spatial and temporal median filtering, and thresholding based on closest match to the treatment field. The images presented correspond to different color scales to optimize dynamic range visibility.

**Table 1 Patient treatment specifications.**

| Patient number | RPO—6X | RPO—10X | LAO—6X | LAO—10X | Prescription dose/total # fractions | Fx's imaged |
|---|---|---|---|---|---|---|
| Pt 1 | ∠ 310° 107 MU | — | ∠ 130° 108 MU | — | 5040 cGy/18 Fx | 5 |
| Pt 2 | ∠ 311° 149 MU | — | ∠ 130° 168 MU | — | 4256 cGy/16 Fx | 10 |
| Pt 3 | ∠ 315° 90 MU | ∠ 315° 11 MU | ∠ 135° 106 MU | ∠ 135° 14 MU | 5040 cGy/28 Fx | 10 |
| Pt 4 | ∠ 310.5° 158 MU | — | ∠ 130.5° 166 MU | — | 4256 cGy/16 Fx | 6 |
| Pt 5 | ∠ 309° 145 MU | — | ∠ 129° 160 MU | — | 4256 cGy/16 Fx | 6 |
| Pt 6 | ∠ 299° 154 MU | — | ∠ 119° 148 MU | — | 4256 cGy/16 Fx | 10 |
| Pt 7 | ∠ 315.5° 50 MU | ∠ 315.5° 52 MU | ∠ 135.5° 55 MU | ∠ 135.5° 55 MU | 5040 cGy/28 Fx | 13 |
| Pt 8 | ∠ 313° 67 MU | ∠ 313° 39 MU | ∠ 133° 67 MU | ∠ 133° 44 MU | 4500 cGy/25 Fx | 17 |
| Pt 9 | ∠ 302° 146 MU | — | ∠ 122° 150 MU | — | 4256 cGy/16 Fx | 11 |
| Pt 10 | ∠ 312° 65 MU | ∠ 312° 84 MU | ∠ 131° 62 MU | ∠ 131° 103 MU | 4256 cGy/16 Fx | 8 |
| Pt 11 | ∠ 310° 160 MU | — | ∠ 130° 161 MU | — | 4256 cGy/16 Fx | 4 |
| Pt 12 | ∠ 295° 145 MU | — | ∠ 115° 160 MU | — | 4256 cGy/16 Fx | 8 |
| Pt 13 | ∠ 305° 52 MU | ∠ 305° 43 MU | ∠ 125° 60 MU | ∠ 125° 51 MU | 5040 cGy/28 Fx | 2 |

Treatment field specifications are organized below for each patient. Each patient received at least one pair of RPO and LAO 6X beams, additionally Pt's 3, 7, 8 and 10 and 13 would receive two, additional, opposing 10X beams. The total prescription dose is provided in the right column, followed by the number of fractions the total dose was distributed over. In the final column, the number of fractions imaged is noted.

beams at 6 MV (6X) and 10 MV (10X), whereas the full treatment for Patients 1 (Fig. 3a), 2 (Fig. 3b), 4 (Fig. 3d), 5 (Fig. 3e), 6 (Fig. 3f), 9 (Fig. 3i), 11 (Fig. 3k), and 12 (Fig. 3l) involved only the use of 6× beams. Additional patient treatment information can be found in Table 1.

A total of 108 sessions of radiotherapy treatments were successfully imaged from $n = 13$ patients and used to develop the discussed methodology. Using these 108 fractions, a negative linear correlation was identified between dose-normalized Cherenkov intensity and subsurface Hounsfield-scale radiodensity of the breast, where Hounsfield units represent the linear attenuation of the primary X-ray beam through the imaged tissue. Figure 4a, b demonstrates the extent of Cherenkov light emitted per unit dose deposited (Gy) variation due to different CT radiodensity and breast tissue patterns. In Fig. 4a, the CT scan of Patient 3 shows the presence of dense, fibroglandular tissue throughout the majority of the breast. Pictured adjacent are Cherenkov images from ten sequential fractions, normalized with respect to the prescribed breast surface dose from the treatment plan, shown as an intensity map in photons $Gy^{-1}$ (see Supplementary Note 1 and Supplementary Fig. 1 for description of surface dose normalization). In Fig. 4b, a CT scan of Patient 8 shows a contrasting example of predominantly adipose tissue. Consequently, the corresponding dose-normalized Cherenkov images to the right exhibit much higher optical signal as compared those in Fig. 4a. In Fig. 4c, the median Cherenkov intensities $I_{uncorr,D}(E)$, normalized by dose (denoted with subscript D) is plotted with respect to the patient average radiodensity in the subsurface tissue for 6 MV beams. Data in Fig. 4c, d were averaged over all fractions imaged, and grouped by beam energy ($E$). Scattered points are color-coordinated by gantry angle (∠) RPO (blue) or LAO (gray), though in linear fitting, RPO

and LAO Cherenkov intensity medians were averaged into one value for computed statistics.

**Cherenkov emission and radiodensity.** The dose-normalized Cherenkov median intensities, $\bar{I}_{uncorr,D}(E)$, were corrected using a normalization from the fits in Fig. 4c, d,

$$\bar{I}_{corr,D}(E) = \bar{I}_{uncorr,D}(E) \cdot c(HU, E), = \bar{I}_{uncorr,D} \cdot \left[ \frac{m(E) \cdot (-135) + b(E)}{m(E) \cdot HU + b(E)} \right], \tag{1}$$

where $\bar{I}_{uncorr,D}(E)$ is the median intensity of the recorded Cherenkov field in units photons per one Gray of dose delivered ($\gamma \, Gy^{-1}$). HU is the average CT attenuation number expressed in Hounsfield Units for the same patient, and $m(E)$ and $b(E)$ are the slope and intercept derived from each fit. The numerator of the correction factor $c(HU,E)$ is gathered by extrapolating back to the $-135$ HU crossing $[m(E) \cdot (-135) + b(E)]$, an arbitrary correction endpoint chosen at a very low tissue density where human breast tissue is unlikely to extend beyond. Note that HU was only extracted in the superficial 10 mm of tissue (Fig. 4a, b, yellow contour) to correspond to the region relevant to escaping Cherenkov photons. Error bars throughout Fig. 4 and Supplementary Fig. 2 are represented using the root mean square error of $\bar{I}_{uncorr,D}(E)$.

**Calibration for Cherenkov intensity.** By isolating parameters in Eq. (1) that are applied directly to the final images, we can define a simple multiplicative correction factor for HU, $c(HU, E)$, unique for the beam energy $E$ and the patient HU. These corrections equalize the data, which can be found in Supplementary Fig. 2. As the resulting $p$ values indicate, the trend between the

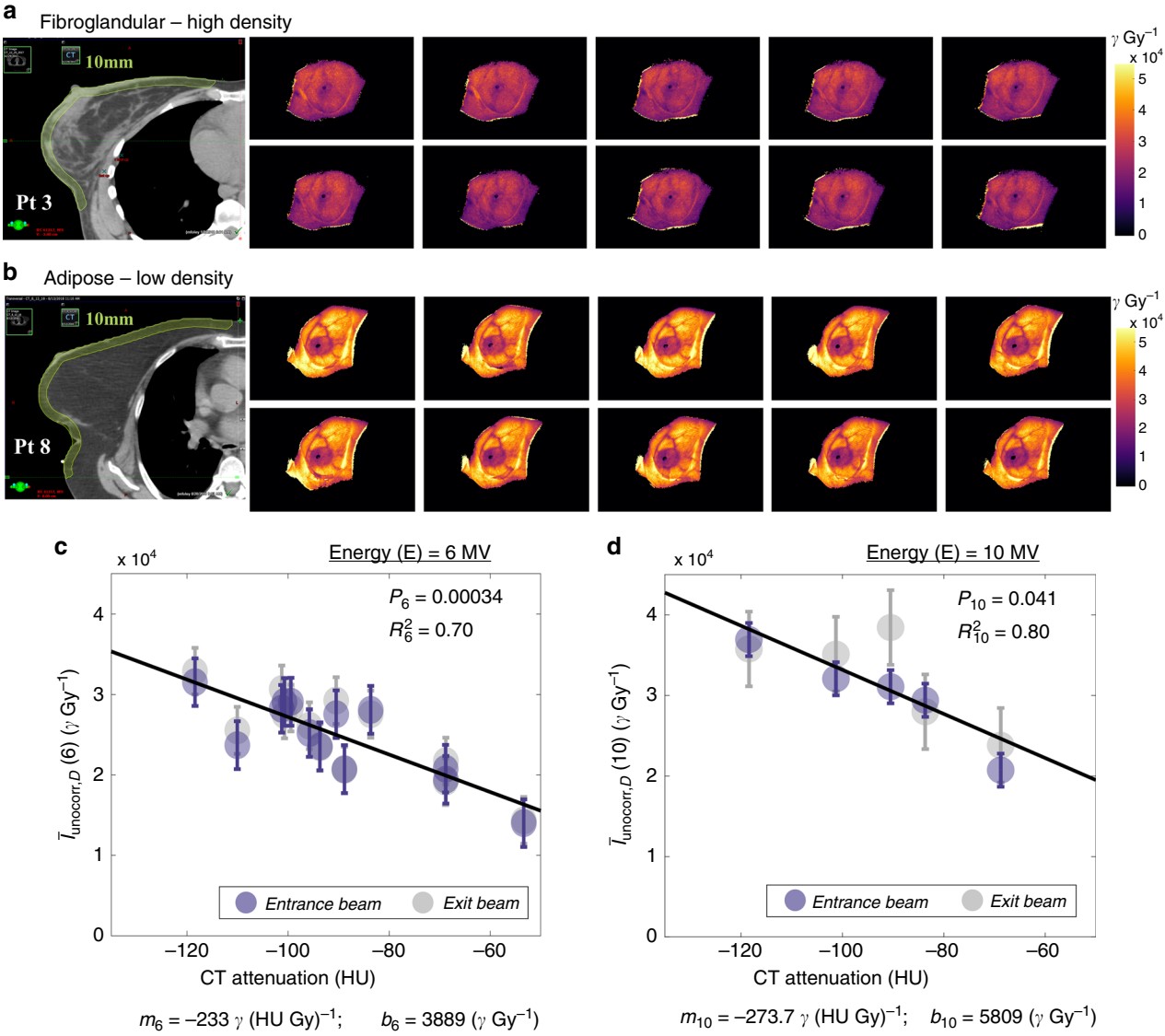

**Fig. 4 Variability in Cherenkov intensity with fibroglandular and adipose tissue content.** In **a**, the CT scan of Patient 3 shows dense, fibroglandular content throughout the breast volume. Ten fractions of Cherenkov imaging are pictured (units photons ($\gamma$)), each normalized with respect to dose (Gy) as shown beside this. In (**b**), the CT scan of Pt 8 is characterized by largely adipose tissue, resulting in a much brighter appearance of Cherenkov images as compared those in **a**. Subfigures **c** and **d** illustrate the linear correlation between HU and the median Cherenkov counts per unit delivered dose ($\gamma$ Gy$^{-1}$), averaged over the number of fractions imaged (mean). Beams are separated by color (Entrance/RPO, gray and Exit/LAO, blue). Subfigure (**c**), $n = 13$, maps this relationship for all 13 patients treated with 6 MV beams, and (**d**), $n = 5$, for the five patients treated with 10X beams. Each point shown is averaged over all fractions for each patient (see Table 1 for number of fractions). RPO and LAO means for each patient were averaged into one point, and the linear correction was computed and applied to each Cherenkov ($\gamma$ Gy$^{-1}$) value for both 6X and 10X beams to normalize the Cherenkov signal by HU value (Supplementary Fig. 2b, d). Error bars shown depict the root mean square error.

CT attenuation of subsurface breast tissue and the amount of Cherenkov light per unit dose deposited was highly significant for 6X beams ($p = 0.0003$), and significant for 10X beams ($p = 0.041$).

The HU-derived scale factors $c$(HU), $E$ can be used to correct the Cherenkov images using the simple expression

$$I_{\text{corr}}(E) = c(\text{HU}, E) \cdot I_{\text{uncorr}}(E),  \quad (2)$$

where each pixel in the uncorrected image matrix $I_{\text{uncorr}}(E)$ is multiplied by the same HU-correction factor for a given energy. The calibrated results are shown in Fig. 5, where the Entrance (RPO) and Exit (LAO) 6 MV beams are shown side-by-side for all 13 patients (Fig. 5a) and organized into columns by the original (uncorrected) Cherenkov image, the surface dose image, and the

HU-corrected Cherenkov image (yielded by the operation in Eq. (2)). As described, these 108 patient data sets (first fraction pictured) were corrected using CT radiodensity within the first 10 mm of tissue, where it becomes observable that the corrected Cherenkov images coincide more closely with the expected surface dose images when restricted to the same relative dynamic ranges. In Fig. 5b, the same is shown, but for LAO and RPO 10 MV beams.

To test the linearity of the relationship between calibrated Cherenkov light and absorbed dose, a final evaluation of the correlation was carried out on all corrected image data sets, testing the same number of regions for each patient across the breast, using all fractions imaged (Fig. 6). Circular ROIs were used to relate uncorrected Cherenkov images $I_{\text{uncorr}}(E)$ to dose ($D$

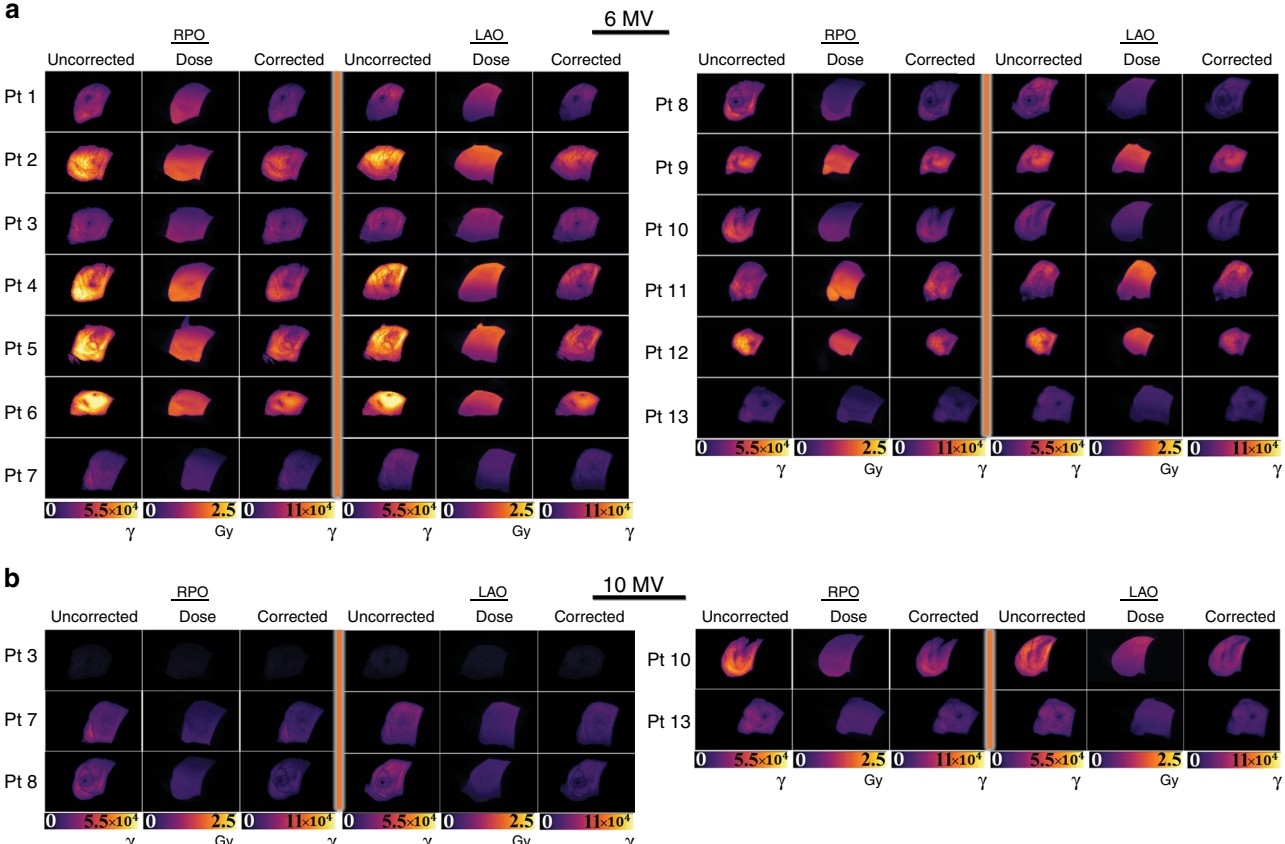

**Fig. 5 Cherenkov images corrected for radiodensity.** The final corrected RPO and LAO images are shown. Subfigure (**a**) organizes the original (labeled Uncorrected) Cherenkov images recorded during 6 MV radiotherapy treatment (units in Cherenkov photons, $\gamma$) as compared to the expected surface dose estimate (labeled Dose) from the treatment plan (Gy). After CT attenuation (HU) corrections have been carried out, the corrected Cherenkov images are shown (labeled Corrected). In (**b**), the same is shown for 10 MV beams. The corrected Cherenkov images qualitatively match the expected surface dose images more closely than the uncorrected images, where further quantitation is presented in Fig. 6.

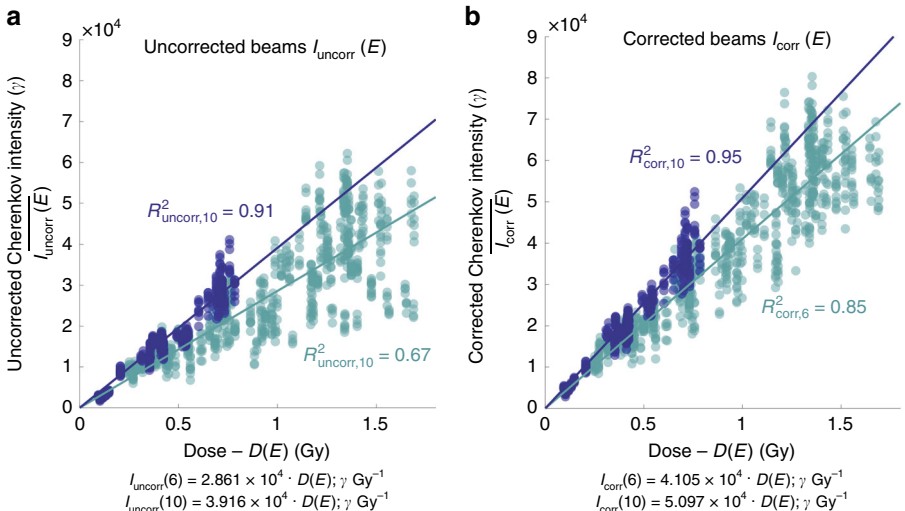

$$I_{uncorr}(6) = 2.861 \times 10^4 \cdot D(E); \gamma \text{ Gy}^{-1}$$
$$I_{uncorr}(10) = 3.916 \times 10^4 \cdot D(E); \gamma \text{ Gy}^{-1}$$

$$I_{corr}(6) = 4.105 \times 10^4 \cdot D(E); \gamma \text{ Gy}^{-1}$$
$$I_{corr}(10) = 5.097 \times 10^4 \cdot D(E); \gamma \text{ Gy}^{-1}$$

**Fig. 6 The HU correction increases linearity between Cherenkov light and dose.** The same number of regions of interest (ROI) per patient are sampled from each corrected and uncorrected Cherenkov image stacks and co-registered surface dose images to yield the relationships shown. The median Cherenkov intensities from uncorrected images (**a**) are plotted against corresponding ROIs from the predicted dose in the treatment plan. The median Cherenkov intensities from the corrected images are then plotted in the adjacent plot (**b**) for both 6 and 10 MV beams. The $y$-intercepts have been constrained to cross at the origin, which yields an uncorrected 6X beam regression $R^2_{uncorr,6} = 0.67$ (light blue), which exhibits the largest spread of data. The HU-corrected 6X beam regression is improved by 0.18, to $R^2_{corr,6}$ of 0.85. The linear regression of the 10X beams are improved slightly by 0.04, from 0.91 to 0.95 (dark blue). The Pearson's linear correlation coefficient increased from $r_{uncorr,6} = 0.82$ uncorrected to $r_{corr,6} = 0.92$ corrected in 6X beams, and from $r_{uncorr,10} = 0.96$ uncorrected to $r_{corr,10} = 0.97$ corrected for the 10X beams. All data are organized by beam energy, where light blue is representative of all 6 MV data and dark blue is representative of all 10 MV data.

$(E)$), whereas corrected Cherenkov image ROIs $I_{corr}(E)$ were first related to the corresponding HU values, then to dose, by a regression fit to a simple linear slope, $m_{corr}(E)$, relating this to dose, $D(E)$,

$$\bar{I}_{corr}(E) = m_{corr}(E) \cdot D(E). \tag{3}$$

Mean values were estimated from these ROIs in the regions of corrected and uncorrected Cherenkov images, avoiding highly absorbing areas (nipple, major vasculature, and scar). These results are plotted in Fig. 6a, uncorrected Cherenkov emission intensity, $\bar{I}_{uncorr}(E)$, and Fig. 6b, corrected Cherenkov emission intensity $\bar{I}_{corr}(E)$. The same ROIs were evaluated for all data sets for a given patient. The $R^2$ linear regression coefficient of the 6 MV beam increased from $R^2_{uncorr,6} = 0.67$ to $R^2_{corr,6} = 0.85$ after correction. The 10 MV Cherenkov to dose relationship metrics also increased from $R^2_{uncorr,10} = 0.91$ to $R^2_{corr,10} = 0.95$ after correction. The Pearson's correlation coefficient was additionally strengthened from $R_{uncorr,6} = 0.82$ to $r_{corr,6} = 0.92$ for 6 MV beams, and minimally from $r_{uncorr,10} = 0.96$ to $r_{corr,10} = 0.97$, for which the correlation was already very strong.

## Discussion

The relationship between emitted Cherenkov light intensity and absorbed dose is linear for monoenergetic beams in homogeneous media such as water or in a single media type; but in all human tissues, the signal intensity is affected by patient-specific optical absorption and scattering attenuation. In this study, the largest cohort of patient images to date was used to examine these parameters, then determine the limits to which they can be linearly corrected for, toward the goal to image absolute dose of the treated field (Fig. 6). This can be verified using tissue phantom studies which show the same trend (Supplementary Fig. 4, Supplementary Note 4). Earlier studies have elucidated the most dominant factors through Monte Carlo simulations[21], finding that tissue composition is one of the largest. The tools used to measure tissue composition for this methodology are readily available in existing radiation oncology practices, as discussed here, including the use of the planning CT simulation for radiodensity values. Past iterations of tissue optical property correction studies have relied upon the use of additional systems and patient contact to determine tissue optical properties. The benefit of using CT radiodensity is clear in radiotherapy, because almost all patients receive a CT scan for treatment simulation in the exact geometry as they are treated, providing a direct map of the treated regions.

Future developments of CT HU correction involve sampling the CT scan at each voxel to yield a spatially-dependent image of CT values. This would serve as a more robust contrast to the scalar HU value used to make tissue corrections in this study, and could correct the Cherenkov images based on spatial variations in the tissue, including for the surgical scar (see Supplementary Note 5)[22].

In this study, data fits were made using patient data from 108 imaging sessions. With increased imaging of additional patients, this hypothesis can be further tested with a wider range of breast densities and (darker) skin types. Recruitment in this study was limited by the demographic of the local catchment area, consisting of primarily Caucasian women. While there are limitations associated with this patient cohort having primarily lighter skin, the optical properties of subcutaneous and deeper tissue does not vary with skin color, so our results show that the methods presented in this study can be applied to all women. Correcting the Cherenkov emission for melanin and other absorbers in the skin are currently being explored, and to date, have been tested in tissue phantoms[21] as well as clinically, though only to limited extents[23]. The future inclusion of patients across a spectrum of skin types and sensitivities to radiation remains to be thoroughly tested.

Breast imaging is a robust example of how to correlate emitted Cherenkov intensities to absolute dose. The two primary breast tissue types are well characterized, with fibroglandular tissue being both radiodense and optically attenuating, and adipose tissue being less radiodense and less optically dense. These same two tissues have differences in blood volume[10,15,24,25,26], which is a primary absorber (if not the most absorbing) of Cherenkov light[21]. While Cherenkov emission provides only surface dose imaging of up to 10 mm depth[6], the CT values for these depths are readily extracted and used for calibration in software. The study completed here avoided major blood veins in the breast, which have been known to be a prominent feature in the Cherenkov images, but given that these are not readily visualized in CT scans, the compensation for their attenuation is not feasible with non-contrast CT image data. Rather, these near-surface veins can be effectively corrected for by simpler optical reflectance imaging methods[16].

This study focuses primarily on the correction of inter-patient differences and the linear dependence of Cherenkov emission relative to dose delivered. In Fig. 6, the raw data show that a range of Cherenkov intensity outputs can be observed for a given predicted dose, indicating notable intra-patient differences exist (illustrated by the ranges of Cherenkov intensities mapped to one dose value, giving the data a stacked appearance). It is suspected that as erythema develops in patients, the attenuation of the Cherenkov light increases, which can be quantified in Supplementary Fig. 3 and discussed in Supplementary Note 3. By taking a reflectance image at the time of treatment, we maintain that surface attenuations such as tattoos or skin color could largely be corrected for by reflectance normalization[16]. Further, to importantly consider breast implant material in post-mastectomy and large lumpectomy patients, a full assessment of these methods awaits testing in this unique patient cohort[27].

The CT attenuation values (HU) of breast tissue correlate linearly with emitted Cherenkov light per unit deposited surface dose, which can be used as an empirical linear calibration factor to obtain absolute non-contact dose imaging. This approach to test for linearity between calibrated Cherenkov and radiation dose was tested in 108 images of radiotherapy treatments from 13 subjects, providing evidence that the approach is valid. In patient treatments when 6 MV beam energies were used, the Cherenkov light to dose linearity was strengthened ($R^2_{uncorr,6} = 0.67$ to $R^2_{corr,6} = 0.85$) as well as in 10X beams ($R^2_{uncorr,10} = 0.91$ to $R^2_{corr,10} = 0.94$). The efficacy of these methods are considerable, provided the utilization of readily available patient CT information that comes at no additional time or monetary expense to the patient or the clinical care team. A reflectance correction may also prove beneficial in further correcting intra-subject variation, which may be integral in the correction of patients with higher skin pigmentation or blood volume variations.

This study presents an approach to non-contact imaging of optical light signal that is linearly correlated to radiation dose in human tissues, offering a relevant step toward absolute radiation dosimetry by non-contact imaging during the treatment procedure.

## Methods

**Human studies**. This human study was approved by the Institutional Review Board (IRB) at Dartmouth College, and all procedures followed the approved protocol. Recruitment for participation in this study included informed consent by each participant for imaging the treatment area during their standard radiotherapy

fractions. All subjects were treated according to their clinically prescribed whole-breast radiotherapy plan (Table 1). All patients (excluding Patients 3, 7, and 8) in this study were prescribed a hypofractionation plan, consisting of five whole-breast (WBR) fractions per week over 4 weeks from an RPO beam and an LAO beam, followed by a boost plan (not imaged). Treatment was delivered by a Varian 2100CD linac (Varian Medical Systems Inc., Palo Alto, CA) and included two tangential 6 MV beams, or two sets of tangential beams at both 6 and 10 MV sequentially. Patients 3, 7, and 8, alternatively, were prescribed a standard-fractionation plans, comprised of a five fraction/week treatment over 6 weeks utilizing four beams centered around the breast, and two super-clavicular fields from different angles. Plan specifications for each field can be found in Table 1.

**Cherenkov acquisition**. A time-gated intensified camera (C-Dose Research, DoseOptics LLC, Lebanon, NH) was used to image the Cherenkov light emitted during radiotherapy over the course of this study. The camera was equipped with a 50 mm f/2.8 lens (Nikon Inc, NY), and mounted on the ceiling at IEC 61217 spatial coordinates of ($x = -1288$, $y = 1066$, $z = -687$) mm from the linac isocenter. The camera image intensifier was enabled only during the duration of each 4 μs X-ray pulse from the linear accelerator at 360 Hz repetition rate, allowing the room light signal to be subtracted at this low duty cycle. The camera uses a stray X-ray detector as a trigger signal to initiate the gating pulse for the intensifier to turn on[16]. Each readout frame from the CMOS camera had a fixed integration time of 51 ms, containing the summed intensity of Cherenkov light from 18 linac pulses. Every Cherenkov readout frame was followed by a background-only readout image, which was achieved by adding a fixed delay of 10 μs after the gating X-ray pulse arrival. The background frames used an 8 ms exposure time and an extended intensifier-on time (720 μs total). Real-time subtraction of the scaled background images from the raw Cherenkov images is performed in the software, providing a video stream of only the radiotherapy field at 17 frames per second. The video stream was temporally filtered using a median filter with 5-frames kernel in order to suppress stray X-ray noise in the images. Each frame was also spatial median filtered using a $5 \times 5$ kernel. As part of camera quality assurance and consistency measures, 200 monitor units (MU) of a $20 \times 20$ cm$^2$ beam was delivered to a white ABS calibration board daily to monitor for any changes in sensitivity. Small intensity variations observed on the calibration board were used to correct patient images prior to analysis.

**Software versions and use**. C-DOSE, used for data acquisition, commercially available, v4.00. Run and tested on Windows 10, Lenovo ThinkPad P51. For use, ensure that your PC meets the minimum specs (16GB ram, 500GB SSD, i7 processor, NVIDIA GTX 1050 Ti or better, Windows 10 Pro 64-bit, USB3). In order to acquire images, a C-Dose camera, optical fiber, and optical repeater will be required to connect to acquisition computer.

MATLAB, used for data processing Commercially Available. Run and tested on Version: R2018A. Currently Operating on Windows 10. No required non-standard hardware.

Image-J (Fiji), Version 1.52a, a public Domain Software by the NIH with free online download was used for general image stack viewing and initial analysis (used to confirm findings found later using MATLAB). This software run and tested on Lenovo ThinkPad P51, operating on Windows 10.

**Statistics and reproducibility**. The Pearson's correlation coefficient and $p$ values were computed using MATLAB function corrcoef. As adapted from ref. [28]: "The $p$-value is computed by transforming the correlation to create a $t$-statistic having $N-2$ degrees of freedom, where $N$ is the number of rows of $X$. The confidence bounds are based on an asymptotic normal distribution of $0.5 * \log((1 + r)/(1 - r))$, with an approximate variance equal to $1/(N - 3)$. These bounds are accurate for large samples when $X$ has a multivariate normal distribution. The 'pairwise' option can produce an r matrix that is not positive definite… Default is 0.05 for 95% confidence intervals."

The Cherenkov intensity from each breast image was sampled in different dose regions within the field of emitted Cherenkov light, and in all cases the standard deviation was less than 10% for each dose-normalized image when high-attenuating regions were avoided. This was repeated and found for all 13 patients in the study. The HU values were sampled from the same estimated regions of the breast depth using an automated thresholding method in Eclipse, to maintain consistency. The correlation between samples was used as our measure of corrections for the Cherenkov intensity.

**Reporting summary**. Further information on research design is available in the Nature Research Reporting Summary linked to this article.

## Data availability
The data that support the findings of this study are available on request from the corresponding author L.A.J. The data are not publicly available for patient privacy purposes. All figures included throughout this work are made using processed data.

## Code availability
MATLAB scripts used in post processing can be found in the following github repository, https://github.com/rachaelhach/NATCOMM_Cherenkov_HU.

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

## Acknowledgements

We are thankful to the radiation therapists at Norris Cotton Cancer Center at Dartmouth Hitchcock Medical Center for their kind assistance during this study, and to the patients who consented to be a part of this diagnostic trial. This work has been primarily funded by NIH grant R01 EB023909, and from R44 CA232879 via equipment loan from DoseOptics LLC, and through use of the Norris Cotton Cancer Center Shared Resources with NCI cancer center support grant 5P30 CA023108-40.

## Author contributions

R.L.H. was responsible for the conception of using CT HU for calibration of Cherenkov light, collection of the images, processing of data, and wrote the paper. M.J. helped design the software and reviewed the data and edited the paper. P.B. helped to design the study, contributed substantially to the development of the analytical techniques, ensured proper camera function, and edited the paper. D.J.G. provided facilities, study design, and expertise in data collection analysis and edited the paper. B.W.P. conceived of study design, provided input on data and algorithm analysis, and edited the paper. The corresponding author L.A.J. was responsible for the clinical study design, patient recruitment and data analysis, and edited the paper. Figure 1 illustration credit to Patricio Saroza (Thayer School of Engineering, Dartmouth College) and R.L.H. (F.A., Thayer School of Engineering, Dartmouth College).

## Competing interests

B.W.P. and L.A.J. have a financial interest in a company, DoseOptics, which manufactures cameras that were used in the current study. This company is funded by SBIR grants. Both authors have a conflict of interest management plan at Dartmouth College and Dartmouth-Hitchcock Medical Center, which includes an independent review of the research integrity prior to publication. Patent disclosure for the work described in this study: X-ray CT Calibration for Quantitative Correction of Cherenkov Light Emission in Radiation Dose Imaging. U.S. Provisional Patent Application No. 62/874,124 on 15 July 2019. Assignee: the trustees of Dartmouth College. Inventors: R.L.H., P.B., B.W.P., L.A.J. Patent 10,201,718. Method and system for using Cherenkov radiation to monitor beam profiles and radiation therapy. Assignee: the trustees of Dartmouth College. Inventors: Pogue; Brian William. Gladstone; David Joseph. Davis; Scott Christian. Axelsoon; Johan Jakob. Glaser; Adam Kenneth. Zhang; Rongxiao. This patent covered preliminary work done prior to this publication for Cherenkov imaging to evaluate radiation treatments. The remaining authors declare no competing interests to disclose.
