## [Peer Review File · Nature Communications]

Reviewers' comments:

Reviewer #1 (Remarks to the Author):

The manuscript represents an important key advancement in using Cherenkov for dose estimation using only imaging modalities. This highly interesting work identifies that CT imaging and computed tissue density can correct differences in Cherenkov imaging and produce a quantifiable Cherenkov image. Overall the manuscript is well assembled but needs further revision to clarify the benefit and novelty of this method. One major claim on line 50 calls out skin color as a major factor for signal attenuation, yet the authors acknowledge patient demographics were limited for this study, thus this model is applicable only to Caucasians at the moment. Furthermore work could be done to improve the model as the axial CT information is averaged to a 10mm depth, yet HU variability can exist in those 10mm (e.g., skin, vs. fat vs. vessels), which could be mapped in greater resolution to the Cherenkov image as imaging angles are known. Lastly vasculature can be a major component in the Cherenkov attenuation, can the CT information help correct for vasculature absorption?

Minor comments include:

- completing the sentence starting on line 75
- checking units are not cut off in figure 4 and 5
- Figure 5 can be transposed for ease of showing each patient and the transformation as outlined in figure 2.
- Comment on Cherenkov variation for patients with surgical scars, and why the patch bolus yielded higher Cherenkov signal
- Can this method account for skin blemishes and or superficial tatoos in the radiotherapy region, what about patients with breast implants?
- Patient 12 should have their arm tatoo deidentified.
- Patient 2 with erythema saw a 300 count decrease, was this decrease cumulative for each fraction, making the Cherenkov losses significant, or did losses over time plateau to a negligible but measureable difference.

Edwin C Pratt & Jan Grimm

Reviewer #2 (Remarks to the Author):

This manuscript discusses a potential practical application of light emitted from the tissue for quantifying a radiation dose in humans via non-contact imaging using the Cherenkov effect. According to the authors, the major limitation of using Cherenkov light comes from the light transport in tissue which can roughly be described with the photon diffusion model. Since the optical parameters of tissue, specifically the absorption and scattering coefficients employed by the diffusion model, are both patient-specific and vary spatially, the linearity between the originating emission within the tissue and the observed signal exiting the skin surface is altered. To overcome this problem, the authors introduce a correction based on a linear model between the dose-normalized Cherenkov intensities and the average X-ray CT attenuation values. The correction appears to be mainly dependent on inter-patient variability, though intra-patient differences could also be taken into account.

Though a significant number of papers on Cherenkov light based superficial tissue dosimetry have been published in the last decade, the rather practical approach presented in this paper is original and, if implemented clinically, could be a significant development. There are several issues, however, which should be clarified before this manuscript could be considered further.

First of all, the authors indicate that the technique could provide only superficial, less than one centimeter below the skin surface, dosimetry information. The relevance of such limited

information on clinical decision-making re: radiation dose should be discussed and clarified in the paper.

Second, the collected Cherenkov light will be heavily weighted toward the superficial tissue layers due to a very significant attenuation of light in tissue. It is not quite clear from the paper how that depth information is taken into account in the employed Cherenkov light transport model. This should be further explained.

Lastly, the notation in the formulas (1)-(4) is rather unclear with many parameters not defined.

In summary, this manuscript suggests a practical way of using Cherenkov light to directly measure a radiation dose inside human tissues without requiring direct patient contact or increasing the time of clinical and post-treatment workflow. The work could be quite useful for clinical medicine and would likely also be of considerable interest to the bioengineering community, though the potential interest to a wider readership audience is less clear. The manuscript could be considered for publication once the authors have addressed the above points.

Reviewer #3 (Remarks to the Author):

You have performed an interesting study showing how CT attenuation values can be used to scale Cherenkov light for determining radiation dosimetry. The technique represents an advance in development of a non-contact method for measuring radiation dose. My major concern is that it does not seem that you have validated the method. If you did a phantom study, or other studies with a corroborative measurement technique this might prove that the attenuation based algorithms that you use do in fact improve the accuracy of quantitation of radiation dose from Cherenkov light.

Additional minor comments:

1. Please make it clear that informed consent was obtained from the subjects and that the IRB approved the study.
2. A limitation of the technique appears to be that it can only estimate the radiation dose for superficial tissues up to 1 cm deep. You might discuss whether or not this is a serious limitation. Does this limit the application to certain applications?

Dear Nature Communications Editorial Committee,

We are resubmitting the manuscript entitled “Imaging Radiation Dose in Breast Radiotherapy by X-ray CT Calibration of Cherenkov Light” to address the points made by all three reviewers. We thank the committee for thorough examination of this work and insightful points made. Reviewer comments are highlighted in yellow, and text added to the revised manuscript are included here in gray.

Response to Reviewer #1

The authors thank the first reviewer for compliments which characterize this work as an important key advancement and for descriptions such as “*The manuscript represents an important key advancement in using Cherenkov for dose estimation using only imaging modalities. This highly interesting work identifies that CT imaging and computed tissue density can correct differences in Cherenkov imaging and produce a quantifiable Cerenkov image.*” Responses to each comment are itemized below.

“*Overall the manuscript is well assembled but needs further revision to clarify the benefit and novelty of this method.*” The authors agree that further revision was needed and discuss the limitations of evaluating dose only at the surface or the tissue. A summary of these details are now highlighted in the introduction in gray highlights. Additionally there are several publications on this work that have previously presented the technology systems used and the nature of the images generated, and so we have referenced these papers rather than overlap too much text on the discussion of the nature of the technique.

“*Furthermore work could be done to improve the model as the axial CT information is averaged to a 10mm depth, yet HU variability can exist in those 10mm (e.g., skin, vs. fat vs. vessels), which could be mapped in greater resolution to the Cerenkov image as imaging angles are known.*” We agree that, at present, we use a single scalar average of the patient HU over the entire treatment field to a depth of 10 mm, and that this can be made more robust in future studies. We chose to average the sub-surface HU values as a reasonable approximation of an “average” tissue type, and due to the linear relationship between optical density and HU, and improving the spatial resolution would likely not affect the main findings

of this work. Using a spatially-resolved correction for optical properties is being explored, but such change would require extensive rework of all data in the paper with expected minimal impact on the presented Cherenkov-dose relationships. In future studies, we propose taking an exponentially weighted surface average at each pixel within the treated region of the image (very similar to the way the treatment plan is sampled to render surface dose images, see Supplementary Figure 1). With a dynamic correction factor matrix for each patient treatment area, we could achieve higher absolute accuracy of this dosimetric method locally on mm-cm scale, rather than per whole irradiated area.

“Lastly vasculature can be a major component in the Cherenkov attenuation, can the CT information help correct for vasculature absorption?” The vasculature was avoided in our analysis, as we were cognizant of this major attenuation from a previous study. The average intensity of Cherenkov was taken from areas without overt surface vasculature, and this is discussed in the Discussions. The CT information in these regions is not that useful, because they do not have contrast injection in simulation CT scans, and so major vessels are not visualized. This remains as a problem to be solved, which is likely much more readily solved by reflectance imaging, as published in our previous paper (Ref 14 in paper). This has been better explained in the paper now in the 2nd paragraph of the Discussions.

“Comment on Cherenkov variation for patients with surgical scars, and why the patch bolus yielded higher Cherenkov signal.”

The patient surgical scar will attenuate the signal at the surgical site, though usually not enough to introduce significant variability to each image. To illustrate this, the median of each Cherenkov image was evaluated again with the surgical site excluded, and the maximum percent difference in intensity recorded for Patient 10 (the largest and deepest surgical scar) was 2.3%. The following has been added to Supplemental Documentation to organize these findings:

Effect of Surgical Scar on Cherenkov Output

The patient treatment scar has the capacity to contribute substantial variability in the treatment field, based on the size of the patient surgical site. Large lumpectomy and partial mastectomy can lead to larger, deeper surgical scars than standard lumpectomy (for which the surgical scar contributes almost no notable effects to the Cherenkov whole-field median intensity). Ultimately, a more sophisticated, pixel-by-pixel type correction method could substantially aid in correcting for surgical scars by producing an image of surface weighted CT values, in contrast to the gross averaging of the most superficial 10 mm of treated tissues, which was done in this study. This would be carried through using a similar sampling as presented

in Supplementary Figure 1: Surface Dose Image Rendering, but used to sample the patient CT scan instead of the patient treatment plan. The method for correction could remain the same, but the input would change from one scalar value to an image of CT surface pixels to accommodate spatial differences in the treated region.

To evaluate a worst-case scenario in the context of this study, Pt 10 (having the largest, deepest, most attenuating surgical scar) images were assessed quantitatively, where it was found between images including and excluding the surgical scar (Supplementary Figure 5), only a 2.3% increase was contributed to the Cherenkov median field intensity. As the technique becomes more refined, this will become a more important consideration. Though with large-scale tissue optical properties contributing to over 40% optical emission variability^[2], these corrections are our first priority and primary concern for this study.

Supplementary Figure 5: Influence of the Surgical Scar on the Cherenkov Intensity. The surgical scar (Pt 10) was masked out to evaluate the Cherenkov intensity with and without the influence of the scar attenuation. Without, an average of 2.28% increase was observed.

“...why the patch bolus yielded higher Cherenkov signal?” Bolus material 1) is made of water, glycerin and acrylic polymer^[1] (brand name: Elasto-Gel) which does not contain natural absorbers like those in the skin, and 2) has been shown in many of our unpublished studies to redirect the Cherenkov light out the edges of the bolus material via light piping. This patch bolus was also placed in the mammary fold, which is a known region of higher Cherenkov emission intensity. While the stark difference in light emission has not yet been compared to differences in dose using a dosimeter, it is reasonably held that these axillary and inframammary folds are in fact receiving a higher dose due to a self-bolusing effect as a result of skin-on-skin folding (Ref. [25] in main article).

“Can this method account for skin blemishes and or superficial tattoos in the radiotherapy region, what about patients with breast implants?” In our previous study [15], we demonstrated the correction of superficial vasculature and absorbers using relatively simple diffuse reflectance imaging. The presented method does not allow us to correct for skin blemishes and superficial tattoos, as its primary objective is to correct for the sub-surface tissue type, and it solely employs the CT maps

as patient-specific correction factor. Final application would ideally combine both approaches, and this research is undergoing. Breast implants are an important consideration for patients having undergone partial or full mastectomy, and have a different HU value and optical attenuation than either fibroglandular or adipose breast tissue. Thus, it is unclear whether implant material could not be accurately characterized by the study here. Thus, future studies should develop a calibration methodology for irradiation of breast tissue with implants, given these known differences in both HU and optical properties. A mention of this has been added in at line 208 of the Discussions section.

“Patient 2 with erythema saw a 300 count decrease, was this decrease cumulative for each fraction, making the Cherenkov losses significant, or did losses over time plateau to a negligible but measurable difference.” The Cherenkov intensity reduction over the course of Patient #2 treatment was averaged to a 316 count decrease per day on the exit side, and a 255 count decrease per day on the entrance side, from a baseline of 28,000 counts (Supplementary Figure 3). This corresponds to about a 1% intensity reduction each day, and after a standard 20 fractions yields a ~20% decrease in signal output due to the development of erythema by the end of treatment. Our data illustrated linearity from beginning to end of treatment, providing no indication of plateau or an erythema-driven signal losses endpoint. This is exactly a motivation for part of the correction, and this is mentioned in the main part of the paper, in the Discussions.

The suggestion that *“Figure 5 can be transposed for ease of showing each patient and the transformation as outlined in Figure 2”* has been carried through and applied. Thank you.

Other minor comments by Reviewer #1, including *“completing the sentence starting on line 75,”* had been fixed, *“checking units are not cut off in figure 4 and 5,”* has been fixed, and *“Patient 12 should have their arm tattoo deidentified”* has importantly been amended as well.

Reviewer #2

The authors thank Reviewer #2 for their opinion that this work holds clinical utility, with comments such as *“the work could be quite useful for clinical medicine and would likely also be of considerable interest to the bioengineering community,”* and additionally that *“Though a significant number of papers on Cherenkov light based superficial tissue dosimetry have*

been published in the last decade, the rather practical approach presented in this paper is original and, if implemented clinically, could be a significant development.”

We directly answer the three concerns raised by this reviewer below:

“First of all, the authors indicate that the technique could provide only superficial, less than one, centimeter below the skin surface, dosimetry information. The relevance of such limited information on clinical decision-making re: radiation dose should be discussed and clarified in the paper.”

Optical light attenuation prevents Cherenkov dosimetry from estimating dose deeper than the surface, thus the authors wish to make it clear that these limitations will be inherent to the methodology. However, it is critical to recognize that while the approach is limited to surface dose imaging, this is the only *in vivo* dose imaging approach ever presented in the history of the use of x-rays in medicine. While there is not volumetric dosimetry here, instead this technique offers the ability to monitor the dose in real time with limited patient or workflow interference. The issue with optical light, while harmless to the patient, is clearly that it is easily absorbed. So, while we certainly acknowledge the limitations permitting only dose estimation at the surface, as mentioned in the paper, the most important feature is to provide real time monitoring of the shape and magnitude of doses *in vivo*. We leave it to the reviewer to decide if imaging of surface dose is more useful than no imaging dosimetry at all. We believe that given the unparalleled nature of this work, that the field should be presented with this unique observation that Cherenkov imaging can be calibrated by CT number to provide accurate surface breast dose *in vivo*.

“Second, the collected Cherenkov light will be heavily weighted toward the superficial tissue layers due to a very significant attenuation of light in tissue. It is not quite clear from the paper how that depth information is taken into account in the employed Cherenkov light transport model. This should be further explained.”

Yes, the Cherenkov light is produced as part of the dose deposition process in media, and so its generation follows a standard depth dose curve for all x-ray radiation. So, the Cherenkov emission intensity actually is expected to quasi-linearly increase within the buildup depth of the photon beam (first 8mm). However, the emission coming out of the skin surface of optical Cherenkov is attenuated by the radiation transport of absorption and elastic scattering, which can be modeled by Monte Carlo or by diffusion theory in the near infrared wavelengths, and largely is an exponential decrease with thickness. Thus, this is a complex topic, and the exact weighting function would need to be modeled with radiation transport studies. Some early Monte Carlo studies from our

group have shown that the signal that is emitted is only originated from the top 10 mm of tissue [Refs 4 & 20] and so we have used this as our approximation of the exiting volume that is sampled by the Cherenkov light. We have ongoing work in this area, but to date, there is not a consensus that one weighting function is appropriate nor even universally applicable. Thus, for the current study we believe that the applied method of averaged HU is appropriate and practical. We have added mention of this to the 3rd last paragraph in the Discussions section of the paper.

“Lastly, the notation in the formulas (1)-(4) is rather unclear with many parameters not defined.” To improve unclear notation and definitions of parameters, we have averaged the two slopes in Supplementary Figure 2(a,c), such that they now appear as they do in the Main Article, Figure 4(c,d), which eliminates the gantry angle dependence (Entrance Beam RPO and Exit Beam LAO). With these two slopes averaged, correction factors now only vary by beam energy, therefore fields may be added, and all image corrections have been coalesced into one Figure, Figure 5). As a result, the equations should appear less cumbersome.

Reviewer #3:

We thank Reviewer #3 for their favorable summary, *“You have performed an interesting study showing how CT attenuation values can be used to scale Cherenkov light for determining radiation dosimetry. The technique represents an advance in development of a non-contact method for measuring radiation dose.”* The only major concern presented by Reviewer #3 is outlined below.

“My major concern is that it does not seem that you have validated the method. If you did a phantom study, or other studies with a corroborative measurement technique this might prove that the attenuation based algorithms that you use do in fact improve the accuracy of quantitation of radiation dose from Cherenkov light.” Yes, we agree with this concern. The hypothesized reason behind the observations here was that the two main tissue types (fibroglandular and adipose), vary in their blood volumes, which varies the optical attenuation, and these tissues also happen to have variations in CT number. Fatty tissue has low blood volume and low CT number, while fibroglandular tissue has high blood volume and high CT number. To test this, a tissue phantom calibration was made to represent this hypothesis *ex vivo*, using bovine lard and bovine (ground beef) tissue in varying proportions from 20% fat up to 90% fat. The results of this phantom work confirmed

that the linear relationship observed for *in vivo* trends could also be observed in a controlled *ex vivo* calibration, and so this has been added as a section to the revised manuscript in the Supplementary Documentation, called *Controlled Calibration: CT Radiodensity and Cherenkov Intensity* (below). This has been a very important addition to the paper which validates the clinical observations.

Controlled Calibration: CT Radiodensity and Cherenkov Intensity

To verify the results seen in patient subjects, a controlled experiment was carried out to correlate the CT radiodensity (HU value) of underlying tissues with the Cherenkov output of those tissues. Phantoms were created from bovine tissues (80% raw muscle tissue and 20% adipose tissue) to mimic fibroglandular content with added adipose tallow (100%, bovine) to create phantoms of varying adipose concentrations (Supplementary Figure 4(a)). The resulting CT values ranged from -173 HU to -12.9 HU, broadly encompassing breast tissue averages seen in our patient cohort. A 30 cm x 30 cm field beam was delivered to the phantoms in the configuration shown, where the treatment plan dose in color wash is shown in (b), which demonstrates even dosing. In (c) the reduction in output is visible with increased denser (fibroglandular-like) content. The correlation is mapped in (d), ($p = 1.9 \times 10^{-5}$, $r = -0.98$) with an $R^2 = 0.96$. Because 100% fat was not mixed the same way that the meat-containing samples did (which mixes in air), it was not consistent with the other samples and was not included in the calibration. Though in Supplementary Figure 4(c) it is clear that fat observably emits more Cherenkov light than does the more radiodense, meat-containing samples. To quantify the light emitted from each phantom and the surface dose in each phantom, circular ROI's were drawn in each image, mean values were taken, and the average Cherenkov intensity output in each phantom was divided by its respective average mean dose for dose normalization. As was observed in patients, the correlation was negative and linear. While this slope is notably less steep than what was observed in patients, we can confidently contribute this discrepancy to the many different factors which affect the tissue optical properties in patients, including primarily differences in skin. Nonetheless, the correlation in Supplementary Figure 4(d) demonstrates that CT number and Cherenkov light emission in radiotherapy is certainly one of these factors.

Supplementary Figure 4: Controlled HU/Cherenkov Calibration. In (a), the phantoms are laid out from most-fat containing to least fat-containing from left to right, top to bottom. In (b), the plan for a 30 cm x 30 cm AP field is shown in the treatment planning system, importantly showing an even dosing of the phantoms (coefficient of variation $\sigma/\mu = 1.9\%$) for the 200 MU delivery. In (c), the Cherenkov image most certainly shows a reduced optical output from phantoms containing increased beef tissue content. In (d), the correlation between the CT radiodensity (HU) and the dose-normalized Cherenkov output is shown with error bars represented by the root mean square error.

To address a minor comment from Reviewer #3, “Please make it clear that informed consent was obtained from the subjects and that the IRB approved the study,” this has been included in the manuscript.

Very similar to the remark presented by Reviewer #1, “a limitation of the technique appears to be that it can only estimate the radiation dose for superficial tissues up to 1 cm deep. You might discuss whether or not this is a serious limitation. Does this limit the application to certain applications?” Yes, we reiterate that optical light penetration through highly absorbing

tissues such as skin will always present an issue. Please see the detailed response to Reviewer #1 on this point, and the associated additions to the manuscript.

We are grateful to the editorial committee at Nature Communications for managing the review of this work, and to the reviewers for their thoughtful comments and questions which have notably improved the paper.

REVIEWERS' COMMENTS:

Reviewer #1 (Remarks to the Author):

All my concerns have been adequately addressed. This is a very interesting paper. Thank you.

Jan Grimm

Reviewer #2 (Remarks to the Author):

The authors have revised and improved the manuscript. I feel that they have also adequately addressed the majority of this reviewer's previous concerns and, therefore, recommend publication.

Reviewer #3 (Remarks to the Author):

I think that you have adequately addressed the points raised by the review and that the manuscript is ready for publication.

Re: NCOMMS-19-33311B: “Imaging Radiation Dose in Breast Radiotherapy by X-ray CT Calibration of Cherenkov Light”

RESPONSE TO REVIEWER COMMENTS

Reviewer #1:

All my concerns have been adequately addressed. This is a very interesting paper. Thank you. We thank the reviewer for their thoughtful comments on the first submission and we are glad that we were able to respond to them accordingly.

Reviewer #2:

The authors have revised and improved the manuscript. I feel that they have also adequately addressed the majority of this reviewer's previous concerns and, therefore, recommend publication. We thank the reviewer for their assessment and are delighted to hear they recommend for publication.

Reviewer #3:

I think that you have adequately addressed the points raised by the review and that the manuscript is ready for publication. We thank the reviewer for their favorable opinion that this study is ready for publication.

We thank the editorial board for their thorough comments and help expediting the publication process. We also thank the reviewers for their positive notes on this study and approval for publication. Please do not hesitate to reach out if anything else is needed.